# LayerDiff: Exploring Text-guided Multi-layered Composable Image Synthesis via Layer-Collaborative Diffusion Model

## Abstract

Despite the success of generating high-quality images given any text prompts by diffusion-based generative models, prior works directly generate the entire images, but cannot provide object-wise manipulation capability. To support wider real applications like professional graphic design and digital artistry, images are frequently created and manipulated in multiple layers to offer greater flexibility and control. Therefore in this paper, we propose a Layer-collaborative diffusion model, named **LayerDiff**, specifically designed for text-guided, multi-layered, composable image synthesis. The composable image consists of a background layer, a set of foreground layers, and associated mask layers for each foreground element. To enable this, LayerDiff introduces a layer-based generation paradigm incorporating multiple layer-collaborative attention modules to capture inter-layer patterns. Specifically, an inter-layer attention module is designed to encourage information exchange and learning between layers, while a text-guided intra-layer attention module incorporates layer-specific prompts to direct the generation of specific content for each layer. Additionally, we introduce a layer-specific prompt-enhanced module to better capture detailed textual cues from the global prompt. We also present a pipeline that integrates existing perceptual and generative models to produce a large dataset of high-quality, text-prompted, multi-layered images. Extensive experiments demonstrate that our LayerDiff model can generate high-quality multi-layered images with performance comparable to conventional whole-image generation methods. Moreover, by supporting both global and layer-specific prompts, LayerDiff enables a broader range of controllable generative applications, including layer-specific image editing and style transfer.

## 1 Introduction

The image generation task from textual descriptions has emerged as a critical research area in the field of computer vision and machine learning, with broad applications spanning graphic design, advertising, and scientific visualization. Among various paradigms, diffusion-based generative models have demonstrated superior efficacy in synthesizing high-qulaity images from text prompts, thereby serving as an invaluable asset for both creative endeavors and practical applications. Nonetheless, a salient limitation of current methodologies (Goodfellow et al., 2020; Nichol & Dhariwal, 2021; Rombach et al., 2022) is their inherent design to generate only single-layer, monolithic images. Such a limitation substantially restricts their applicability and versatility in scenarios that require a greater degree of control and adaptability, particularly in professional graphic design and digital artistry where layered compositions are frequently indispensable. In these applications, images are rarely created as a single piece; instead, they are composed and manipulated in multiple layers that allow for iterative refinement, fine-grained adjustment, and object-specific manipulations. Therefore, in this paper, we concentrate on building a layer-collaborative diffusion-based framework and achieving the multi-layered, composable image synthesis task.

Several approaches to layered-image synthesis exist, such as the Text2Layer (Zhang et al., 2023), which employs an autoencoder architecture and a latent diffusion mechanism for dual-layer image synthesis. However, Text2Layer confines its capabilities to only two-layered compositions, thus not fully capitalizing on the expressive power of multi-layered synthesis. Another potential approach can be a post-processing pipeline involving word tokenization (Honnibal et al., 2020),

perceptual segmentation (Kirillov et al., 2023), and inpainting techniques (Rombach et al., 2022). Although functional, this post-processing strategy introduces considerable computational overhead and is prone to error accumulation through multiple processing stages, potentially leading to content or stylistic inconsistencies. Hence, the central aim of this study is to utilize a unified diffusion model framework for accomplishing end-to-end multi-layer image synthesis.

In this paper, we introduce a novel layer-collaborative diffusion model, termed **LayerDiff**, which is meticulously engineered to facilitate text-guided, multi-layered, and composable image synthesis. In contrast to conventional whole-image generative models, LayerDiff pioneers a new paradigm in layer-based image generation, offering the ability to synthesize images comprising multiple constituent layers. The resulting composite image is structured into a background layer, an ensemble of foreground layers, and individual mask layers that define the spatial arrangement of each foreground element. To achieve fine-grained control over the number of layers and the content within each layer, we introduce layer-specific prompts in addition to the global prompt that governs the overall image content. We present the layer-specific prompt enhancer designed to extract intricate textual features and object relationships from the global prompt. Furthermore, we introduce the layer-collaborative attention block that facilitates inter-layer interaction and layer-specific content modulation through the employment of an inter-layer attention module and a text-guided intra-layer attention module.

To construct a dataset suitable for training LayerDiff, we introduce a carefully designed data acquisition pipeline aimed at generating high-quality, multi-layered composable images. Fisrtly, we utilize open-source tools for image captioning to obtain a broad overview and specific details of the images. The layer-specific prompts are parsed using natural language processing techniques, and then we employ state-of-the-art open-vocabulary perception algorithms for object localization of each prompt. Finally, we extract the elements corresponding to each prompt to form multiple layers, and employ inpainting models to fill in the missing portions at the respective locations after extraction. Comprehensive experiments confirm that our LayerDiff architecture excels in producing high-fidelity multi-layered images, exhibiting performance metrics on par with traditional full-image generative techniques. Additionally, LayerDiff's support for both global and layer-specific prompts expands its utility across a diverse set of controllable generative tasks, such as layer-wise composable image manipulation and style transfer.

To summarize the contribution:

- We introduce LayerDiff, a layer-collaborative diffusion model which employs layer-collaborative attention blocks for inter- and intra-layer information exchange. A layer-specific prompt-enhanced module further refines content generation by leveraging global textual cues.

- We introduce a streamlined data acquisition pipeline that generates multi-layered composable images for LayerDiff, integrating state-of-the-art techniques in image captioning, object localization, segmentation and inpainting.

- Our LayerDiff architecture not only generates high-fidelity multi-layered images with performance comparable to traditional whole-image generation methods but also offers versatile control for various generative applications.

## 2 RELATED WORK

**Text-to-Image Synthesis.** The image generation models can be broadly categorized into several architectures, including VAE (Huang et al., 2018), flow-based (Ho et al., 2019), GAN (Brock et al., 2018; Kang et al., 2023), autoregressive model (Yu et al., 2022; Wu et al., 2022) and diffusion models (Nichol & Dhariwal, 2021). Leveraging the advantages of diffusion models in the field of image generation, many recent works Saharia et al. (2022); Gu et al. (2022); Rombach et al. (2022) have started using large-scale text-image pairs for training models, achieving remarkable performance in the text-to-image generation domain. One of the most representative works is Stable Diffusion (Rombach et al., 2022), which applies diffusion to the latent image encoded by an autoencoder, and employs a UNet-like architecture to learn the denoising task. It utilizes a text encoder, pretrained with visually aligned text from CLIP Radford et al. (2021), to encode textual descriptions and injects textual guidance into the network in the form of cross-attention. Text2Layer (Zhang et al., 2023) proposes a layered image generation model, dividing an image into a foreground image, a foreground mask and a background image and encoding layer information into a latent space

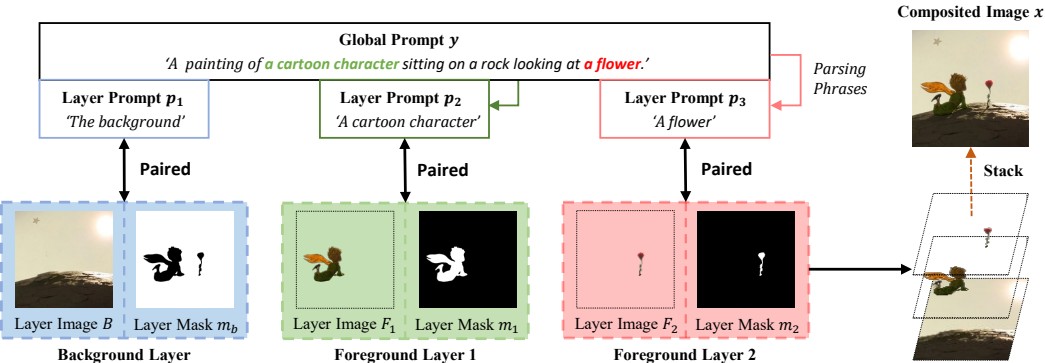

Figure 1: Examples of a Multi-layered Composable Image generated by our data construction pipeline.Based on the global prompt associated with the image, we employ a phrase parsing approach to extract the specific content phrase for each individual layer. Subsequently, we generate the corresponding layer images and layer masks based on their respective layer prompts. By assembling these layers according to the positional information provided by the masks, we are able to construct the final composite image.

using an autoencoder, with layer generation accomplished through a latent diffusion model. Distinguishing from the aforementioned models, we firstly introduce the task of multi-layered composable image synthesis. LayerDiff can generate two or more layers and introduces layer-specific prompts to finely control the content generation for each layer.

**Controllable Image Synthesis.** Various approaches have explored to utilize many control signals to guide image generation. Building upon Stable Diffusion, through fine-tuning text embedding, Textual Inversion (Gal et al., 2022) and DreamBooth (Ruiz et al., 2023) enables the generation of personalized objects in novel scenarios. P2P (Hertz et al., 2022) and PnP (Tumanyan et al., 2023) introduce attention mechanisms with cross-attention to achieves for text-guided content editing. The layout-to-image generation methods (Xue et al., 2023; Zheng et al., 2023) aims to incorporates complex scene information in the form of layouts to achieve more precise scene control. Control-Net (Zhang et al.) injects multiple modal control conditions into the model as images, enabling precise control over object structure in image generation. The control modalities in ControlNet encompass Canny edge maps, semantic maps, scribble maps, human poses, and depth maps. In contrast to the aforementioned models, LayerDiff does not employ complex control signal but rather relies on simple text-based control. Moreover, it introduces layer-specific prompts to achieve precise control over layer content generation. After pre-training, our LayerDiff can directly facilitate multi-level control applications without any fine-tuning.

## 3 METHOD

In this section, we first define the data format of the multi-layered composable image and the task formulation to achieve multi-layered composable image synthesis. Then, we introduce the network architecture of LayerDiff. The layer-collaborative attention block is proposed to learn inter-layer relationships and generate the intra-layer content guided by layer-specific prompts and the layer-specific prompt enhancer is employed to further enhance the controllability of layer-specific elements. Lastly, we outline the pipeline for constructing the training dataset.

### 3.1 TASK FORMULATION

#### 3.1.1 DEFINITION OF MULTI-LAYERED COMPOSABLE IMAGE

Compared to one entire image, a composable image $x$ is composed of multiple layers $L = \{L_i\}_0^k$ a background layer $L_0$ and $k$ number of foreground layers $\{L_i\}_1^k$. A foreground layer $L_i$ is a pair of foreground image $F_i$ and foreground mask $m_i$. There is no overlap between the foreground images and foreground masks, respectively. The foreground mask indicates the area of the foreground object and the foreground image depicts the specific foreground object in this area. The background layer $L_0$ also includes a background image $B$ and a background mask $m_b$. The background mask is actually obtained by taking the complement of the foreground masks. Those foreground pairs are

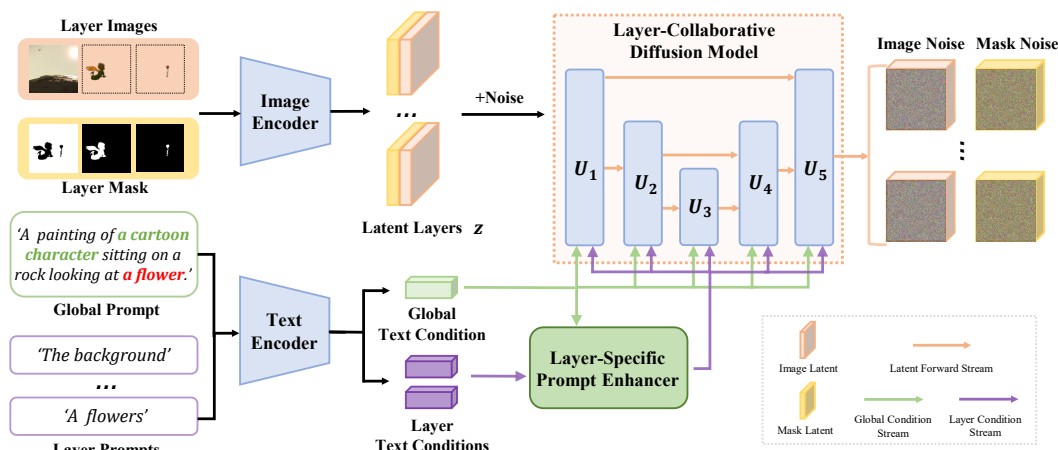

Figure 2: Overall architecture of the proposed LayerDiff. LayerDiff employs an image encoder to extract visual latent for both the layer images and layer masks, as well as a text encoder dedicated to extracting textual features for the global prompt and layer prompts. Subsequently, these layer-specific prompt features are augmented via a Layer-Specific Prompt Enhancer using the global prompt features. Finally, the enhanced layer prompt features and global prompt features are fed into a layer-collaborative diffusion model to predict image noise and mask noise.

arranged along the depth information so that they can be stacked from top to bottom and overlaid onto the background image to compose an entire image. Therefore, a multi-layered composable image $x$ can be defined as:

$$x = m_b * B + \sum_{i=1}^{k} (m_i * F_i) \qquad (1)$$

where $m_b = 1 - \sum_{i=1}^{k} m_i$.

### 3.1.2 TEXT-GUIDED MULTI-LAYERED COMPOSABLE IMAGE SYNTHESIS.

Unlike previous text-to-image generation tasks, which required models to generate images guided by a single comprehensive image description, multi-layered composable image synthesis not only demands a global text description to depict the overall image content but also introduces layer-specific prompts to guide the generation of specific foreground elements in each layer. Furthermore, the layer-specific prompts can provide more detailed descriptions of the foreground objects. We use the layer-specific prompt of "the background" for the background layer and the object phrases as the layer-specific prompts for the foreground layers.

We follow the standard diffusion models to formulate the text-guided multi-layered composable image synthesis. We perform the diffusion process on the composition of multiple layers to add $T$ steps random Guassian noies $\epsilon$ to gradually convert the original multi-layered image $x$ into a random Gauassian distribution $x_T$. LayerDiff is trained to predict the different level of noise on all layers in the opposite direction of the diffusion process. Following the LDM (Rombach et al., 2022), we can apply the diffusion process on the latent space of the layer images $\{z_i\}_0^k$ and layer masks $\{z_i'\}_0^k$ by utilize its latent encoders respectively. In our LayerDiff, we also handle the layer masks in the RGB space by simply repeating one channel to three channel so that the layer mask encoder $\mathcal{V}$ and decoder $\mathcal{D}$ are the same as those for the layer images. Therefore, for a composable image $x$ the diffusion-based multi-layerd composable image generation loss $\mathcal{L}$ under the guidance of the global prompt $y$ and layer-specific prompts $p = \{p_0, ..., p_k\}$ can be formulated as:

$$\mathcal{L} = \mathbb{E}_{\mathcal{V}(x), \epsilon \sim \mathcal{N}(0,I), t} \left[ \|\epsilon_z - \Phi_u(\{z_i\}_0^k, \{z_i'\}_0^k, t, \mathcal{T}(y), \{\mathcal{T}(p)\}_0^k)\|^2 \right] \qquad (2)$$

where $\mathcal{T}$ is the text encoder for the global prompt and layer-specific prompts.

In the sampling stage, LayerDiff generates all layer image latents and layer mask latents by gradually removing the noise from a random Gaussian noise signal over a finite number of steps. We can apply the latent decoder $\mathcal{D}$ to decode back the layer images and layer masks into the original data space.

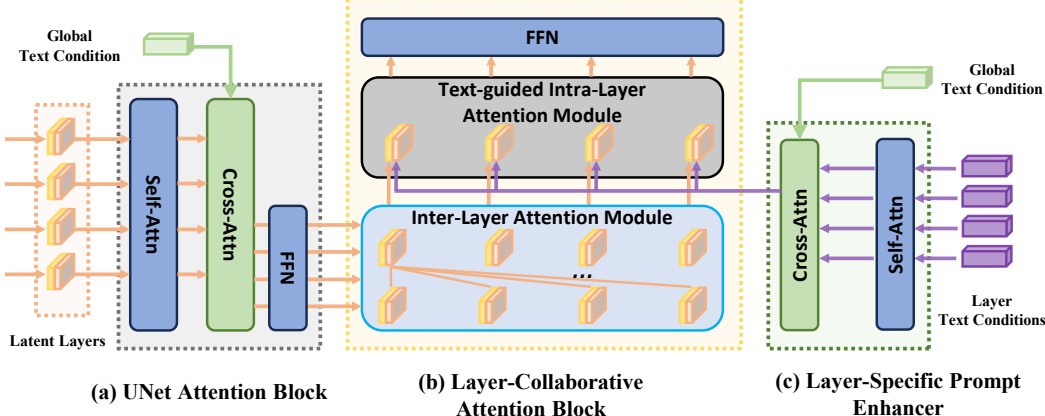

Figure 3: Detailed structure of the proposed layer-collaborative attention block. (a) The UNet attention block is commonly used in the traditional UNet Block for condition-based whole-image generation. (b) To better capture layer-wise features, we introduce the Layer-Collaborative Attention Block, which incorporates a Text-guided Intra-Layer Attention Module to guide layer content generation and an Inter-Layer Attention Module to enable cross-layer interaction. (c) The Layer-Specific Prompt Enhancer is designed to more effectively allow layer-specific prompts to assimilate information from the global prompt.

## 3.2 LAYER-COLLABORATIVE DIFFUSION MODEL

### 3.2.1 ARCHITECTURE OVERVIEW

As shown in Fig. 2, Our LayerDiff is designed to include the image encoder, text encoder, layer-specific prompt enhancer and a layer-collaborative diffusion model. The image encoder $\mathcal{V}$ is employed to convert the layer images and layer masks from the RGB space into the latent space. Note that we treat the layer mask as the RGB image by repeating one channel into three channel. A text encoder $\mathcal{T}$ is applied for the global prompt $c$ and layer-specific prompts $p$ to obtain the global text condition $\mathcal{T}(y)$ and layer text conditions $\{\mathcal{T}(p_i)\}$, respectively. The layer-specific prompt enhancer is proposed to ensure the information completion and controllability of each layer prompt. Both the global text condition and the enhanced layer text conditions are utilized to guide the generation of multi-layered composable images in the layer-collaborative diffusion model, controlling the overall content generation as well as the content generation for individual layers. The layer-collaborative diffusion model draws inspiration from the network design in Stable Diffusion (Rombach et al., 2022). Following the original attention block performing the guidance of the global prompt, our LayerDiff introduces the layer-collaborative attention block, details in Sec. 3.2.2, to learn inter-layer connections and guide the generation of content for individual layers. For the input of the layer-collaborative diffusion model, we concatenate the latent image and latent mask along the channel dimension and stack all layer latents in the layer dimension.

### 3.2.2 LAYER-COLLABORATIVE ATTENTION BLOCK

Layer-collaborative attention block serves as a pivotal component in the multi-layered composable image synthesis, orchestrating the intricate interplay between different layers and guiding the generation of layer-specific content. As shown in Fig. 3, the layer-collaborative attention block is structurally composed of the inter-layer attention module, text-guided intra-layer attention module and a feed-forward network (FFN). The inter-layer attention module is dedicated to learning across different layers. It processes each pixel value of the layer hidden states, capturing the relationships and dependencies between layers, ensuring that the synthesized image maintains coherence and harmony across its depth. The text-guided intra-layer attention module takes the helm when it comes to layer-specific content generation. Guided by the layer text conditions, it ensures that each layer of the image is generated in alignment with the specific textual descriptions, allowing for precise and contextually relevant layered image synthesis. The FFN further process and refine the outputs from the attention modules.

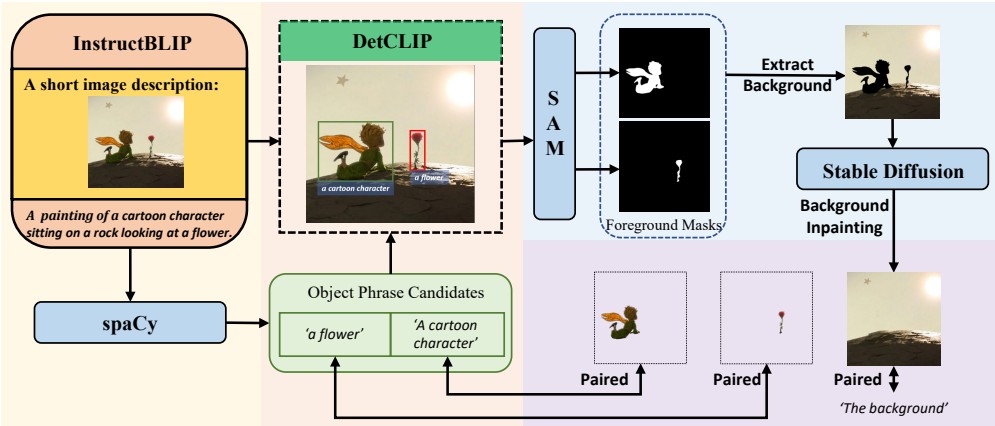

Figure 4: Pipeline of the Multi-Layered Composable Image Construction. We use the InstructBLIP for image captioning and extract noun phrases with spaCy as layer prompts. These prompts guide open-set segmentation via DetCLIP+SAM to produce image layers and masks and the background image is refined by using the Stable Diffusion inpainting model.

### 3.2.3 LAYER-SPECIFIC PROMPT ENHANCER

Layer-specific prompt enhancer is a module designed to refine and augment layer-specific prompts by extracting and integrating relevant information from the global prompt. It potentially ensure more accurate and detailed guidance for individual layer generation in a multi-layered synthesis process. As shown in Fig. 3, a self-attention layer is firstly applied to layer-specific prompts to enhance their distinctiveness from each other, further ensuring the independence of content between layers. Additionally, a cross-attention layer simultaneously takes layer-specific prompts as queries, global prompts as keys and values, aiming to enable layer-specific prompts to capture richer contextual information from the global prompt and inter-layer object relationships.

### 3.3 DATASET CONSTRUCTION

In Figure 4, we propose a data construction pipeline to construct our Multi-Layered Composable Image Dataset, named MLCID. Initially, we employ InstructBLIP (Dai et al., 2023) to generate a caption as the global prompt that can provide a precise and contextually-rich description of the image content for each image. Then, in order to extract a set of foreground images, we apply spaCy (Honnibal et al., 2020) to parse object phrases from the global prompt. With the object phrases candidates, DetCLIP (Yao et al., 2022) or others open-vocabulary detector (Liu et al., 2023) can locate the foreground object in the image and generate its bounding boxes and extract the boxes's label name as the layer prompts, which can be further provided to SAM (Kirillov et al., 2023) to obtain a set of foreground masks corresponding to the layer prompts. Simultaneously, the background masks can be derived through the complementary calculation. Furthermore, the layered images contain one background image and a set of foreground images can be extracted from the image by element-wise multiplication between the image and masks. The decomposition of the image into multiple layers introduces the hollow regions in the background image, which could potentially impact the quality of image generation during training. To mitigate this challenge, we employ Stable Diffusion inpainting model (Rombach et al., 2022) to inpaint the background image. Finally, for each image, we construct a training sample consisting of global prompt, layer images with their corresponding layer prompts and masks. Details of the pipeline can be found in Appendix.

By using the proposed data construction pipeline, we obtain the high-quality, multi-layered, composable image dataset, named MLCID. We collect the training set including 0.8M data from the LAION400M dataset (Schuhmann et al., 2021) and an additional 0.3M data from the private dataset. Furthermore, from the LAION dataset, we extracted 500 dual-layer samples, 500 tri-layer samples, and 200 quad-layer samples to evaluate the model's performance in two-layer, three-layer, and four-layer multi-layer composable image generation, respectively. To ensure a more accurate evaluation, we manually revised the prompts within the LAION test dataset. Additionally, we randomly selected 1,000 samples from a private dataset to include in the test set, resulting a 2.2k testing set in total.

| | Two Layers | | Three Layers | | Four Layers | | All-2.2k | |
|---|---|---|---|---|---|---|---|---|
| | FID | CLIP-Score | FID | CLIP-Score | FID | CLIP-Score | FID | CLIP-Score |
| Stable Diffusion | 105.1 | 31.8 | 110.6 | 31.5 | 166.0 | 31.2 | 42.4 | 31.3 |
| LayerDiff | 102.2 | 31.0 | 121.6 | 29.0 | 187.6 | 27.5 | 58.3 | 30.3 |

Table 1: Main result on text-to-image synthesis on MLCID. Ours LayerDiff can perform multi-layered composable image synthesis. LayerDiff can achieve the text-to-image synthesis by composing the multi layers into one composite image.

| Layer-Specific Prompt Enhancer | Layer Prompt | Mask Dilate | Two Layers | | Three Layers | | Four Layers | |
|---|---|---|---|---|---|---|---|---|
| | | | FID | CLIP-Score | FID | CLIP-Score | FID | CLIP-Score |
| None | ✗ | 5 | 200.9 | 22.6 | 236.7 | 20.4 | 289.2 | 19.4 |
| None | ✓ | 0 | 113.2 | 29.9 | 135.9 | 28.3 | 195.8 | 27.3 |
| None | ✓ | 5 | 105.3 | 30.3 | 127.8 | 28.8 | 192.1 | 27.1 |
| $\mathcal{S}$ | ✓ | 5 | 107.7 | 30.5 | 125.6 | 28.8 | 190.4 | 27.4 |
| $\mathcal{S} + \mathcal{C} + \mathcal{F}$ | ✓ | 5 | 106.8 | **30.8** | 145.5 | 28.0 | 213.0 | 26.5 |
| $\mathcal{S} + \mathcal{C}$ | ✓ | 5 | **103.7** | 30.6 | **123.1** | **29.2** | **182.8** | **28.1** |

Table 2: Ablation study on the components. $\mathcal{S}$ means the self-attention layer. $\mathcal{C}$ is the cross-attention layer. $\mathcal{F}$ is the Feed-Forward Nerual Network.

# 4 EXPERIMENTS

In this section, we first describe the implementation details of our LayerDiff (Sec. 4.1) and experimental details (Sec. 4.2). In our quantitative results (Sec. 4.3), we compare the Stable Diffusion about the text-to-image synthesis on private benchmark and public benchmark and conduct ablation study of proposed components. In the quantitative results 4.4, we visualize the quality of the multi-layered composable image synthesis and explore the application of LayerDiff, including layer inpainting and layer style transfer.

## 4.1 IMPLEMENTATION DETAILS

To expedite the convergence of our model, we initialized our autoencoder and 3D U-Net using the weights from Stable Diffusion v1.5 (Rombach et al., 2022). The detailed parameters initialization rule can be found in Appendix. During the training process, we have a 50% probability of applying the same timestep $t$ to all layers, and a 50% probability of applying different timesteps $t$ to each layer. To prevent a decline in generation performance due to the overly tight segmentation of foreground objects by the masks, we employ a dilation operation on the masks when obtaining foreground images. The dilation kernel of size is $5 \times 5$. However, during training, the input masks are used without the dilation operation.

## 4.2 EXPERIMENTAL DETAILS

We pre-train models on the MLCID with about 1.1M multi-layered image-text pairs. The resolution of the image is $256 \times 256$. Following Stable Diffusion, the pixel values of the layer images and layer masks are all normalized to [-1, 1] and the autoencoder and text encoder are frozen. We apply the AdamW optimizer with a weight decay of 1e-4. The learning rate is set as 1e-5 with 500 steps linear warmup and kept unchanged until the training is finished. The batch size is set as 32 and we apply 8 steps gradient accumulation to stabilize the training and obtain better generalization performance. We randomly drop 10% global text condition and layer text condition in our pre-training.

In our sampling stage, we apply DDIM Scheduler () with 100 iterations and classifier-free guidance (Ho & Salimans, 2022) scale of 3. Specifically, instead of using the empty text as the unconditional text, we use the concatenation of the foreground layer texts as the negative global text condition and negative layer text condition for the background layer. Similarly, we use the background prompt, i.e., 'the background' as the negative text condition for the foreground layer.

Evaluation: We evaluate our LayerDiff on the MLCID's testing dataset with 2.2 k samples. We use the open-source tool, torch-fidelity (Obukhov et al., 2020), to evaluate the metrics of FID. We also apply the CLIP-score to evaluate the alignment between generated composable images and the given prompts by using the RN50 backbone of CLIP (Radford et al., 2021). We compare to Stable Diffusion (Rombach et al., 2022) by directing generating images of a resolution of 512 and then resizing them to 256.

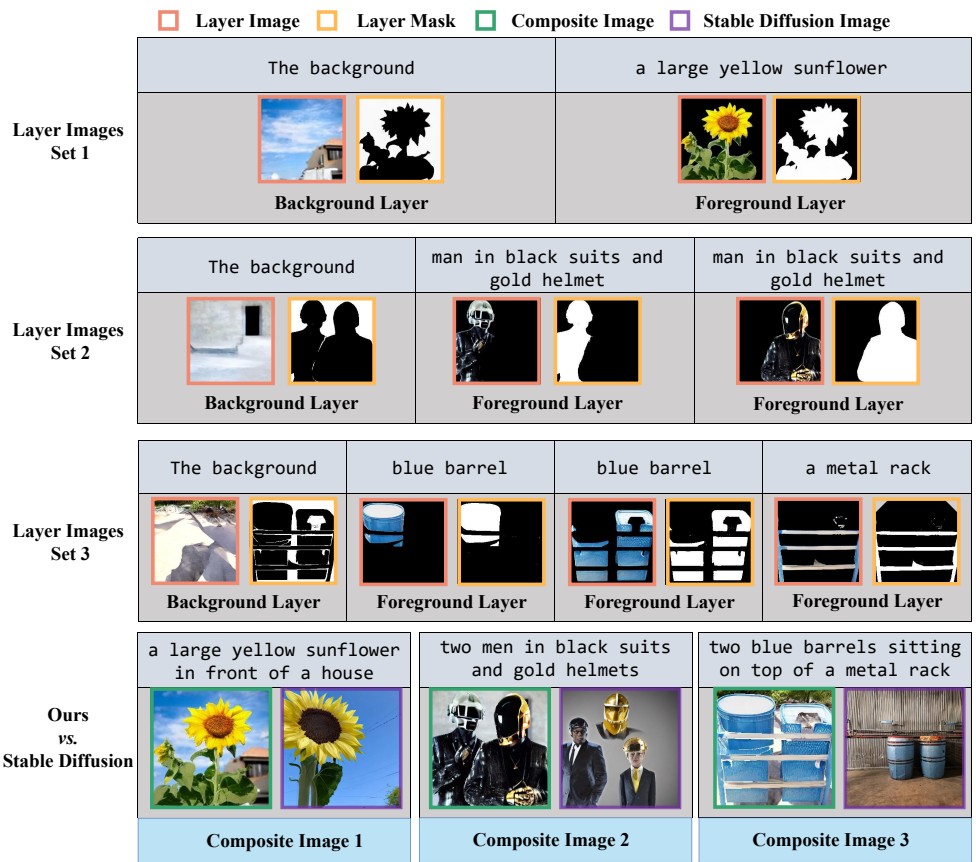

Figure 5: Visualization of the synthesized multi-layered composable images. We present the results of two-layer, three-layer, and four-layer generation, including layer images and layer masks. We compare our composited image with the samples generated by Stable Diffusion using the whole-image generation approach. The quality of our multi-layer generation is found to be comparable to the samples produced by Stable Diffusion.

## 4.3 QUANTITATIVE RESULTS

**Main Results.** Table 1 demonstrate our main results on the MLCID. Our LayerDiff's performance on two-layered image synthesis is comparable to Stable Diffusion's whole-image synthesis performance. In tests involving a greater number of layers and the full-test set of MLCID, we observed a performance disparity in LayerDiff. We speculate that this gap stems from the insufficient training of our model on three and four layers. Within our MLCID, the representation of three-layer images and four-layered images are merely around 20% and 5%, respectively. This inadequacy in the data volume for three-layer and four-layer images hinders the model's performance in generating images with these specific layer counts. In the future, we aim to further enhance the generalization ability of our model by enlarging the scale of the pre-training dataset.

**Ablation Study.** We conduct the ablation study on a subset with 0.2M data of the MLCID and all models are pre-trained about 50k iterations. From the result of ablation study on Table 2, it is discerned that the layer-specific prompt plays a crucial role in multi-layer generation. The layer-specific prompts assists in guiding the model to generate content for each layer and establish inter-layer relationships. Furthermore, the layer-specific prompt enhancer, utilizing only self-attention and cross-attention layers, can sufficiently augment the information within the layer-specific prompt and improve generation performance, especially in the synthesis of images with more than two-layer. Additionally, employing dilated layer masks to obtain layer images proves to be beneficial for multi-layer generation.

## 4.4 QUALITATIVE RESULTS

Figure 5 displays the layer generation in our model and visualizes the produced individual layer images, layer masks, along with the final results in comparison with those obtained via Stable Diffusion. Through visualization, it is evident that our LayerDiff demonstrates the capability to generate multi-layered composable images. By overlaying and composing the layer images and image

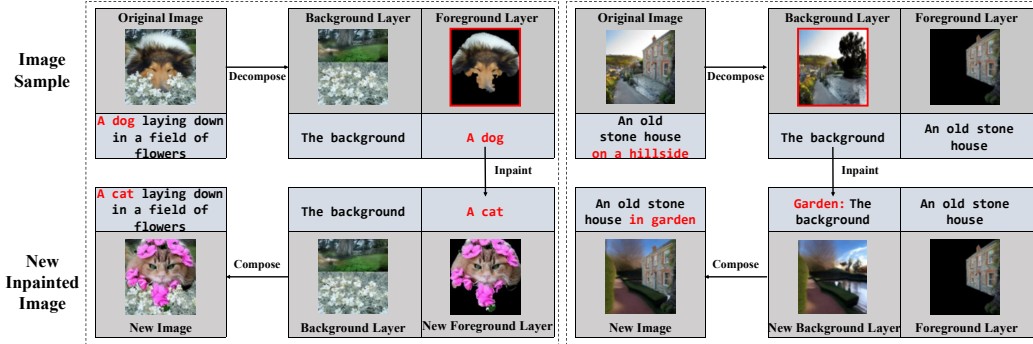

**Figure 6:** Visualization of layer inpainting. We demonstrate the foreground layer inpainting in (a) and background layer inpainting in (b).Our LayerDiff supports the layer inpainting without any fine-tuning. Layer inpainting can strictly control foreground generation and background layer generation within the mask area.

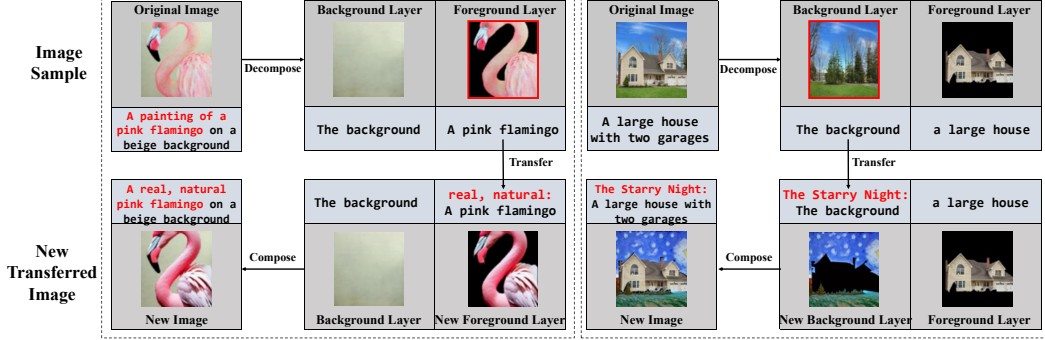

**Figure 7:** Visualization of layer style transfer. We demonstrate the foreground layer style transfer in (a) and background layer style transfer in (b). Our LayerDiff supports the layer style transfer without any fine-tuning. LayerDiff can easily translate the specific layers into the wanted style.

masks, the composite images generated by LayerDiff achieves competitive results when compared to the whole-image generation method, namely, Stable Diffusion.

### 4.5 APPLICATION

After pre-training, our model naturally supports various applications without any fine-tuning.

**Layer Inpainting.** By fixing certain layers and modifying the layer prompt of a specific layer, we can facilitate the regeneration of that layer from noise. Figure 6 illustrates the effects of our background layer inpainting and foreground layer inpainting. Concurrently, LayerDiff's layer inpainting strictly confines the generation of objects within the target layer area.

**Layer Style Transfer.** We can achieve layer-level style transfer. By introducing a certain degree of noise to a specific layer and appending stylized text to the original layer prompt, we instruct the model to perform denoising sampling for the specified layer, thereby accomplishing style transfer for that layer. For instance, as depicted in Fig. 7, we can induce a Van Gogh style transformation for the background by specifying the style through the layer prompt.

### 5 CONCLUSIONS

In conclusion, we introduce LayerDiff, a model designed for text-guided, multi-layered image synthesis. Unlike traditional whole-image generative models, LayerDiff enables object-wise manipulation by leveraging layer-collaborative attention modules. Empirical results show that LayerDiff achieves comparable image quality while offering greater flexibility and control. This work not only extends the capability of text-driven generative models but also paves the way for future advancements in controllable generative applications, including layer-specific editing and style transfer.

**Limitation.** Existing multi-layer training data generation pipelines are inefficient, hindering the capability to produce large-scale training data effectively. This inefficiency compromises the performance of the resulting models, preventing them from achieving impressive results. Therefore,

optimizing the design of efficient multi-layer training data pipeline represent promising future research directions.

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

# A APPENDIX

## A.1 IMPLEMENTATION DETAILS

We initialize our model parameters by utilizing the Stable Diffusion. We random initialize the inter-layer attention module and the layer-specific prompt enhancer. The text-guided intra-layer attention module is initialized by the original cross-attention layer in Stable Diffusion. To make sure the effectiveness of the parameters initialized from Stable Diffusion, we zero-initialize the weight of the last linear layer of each module in the layer-collaborative attention block and layer-specific prompt enhancer. Additionally, We initialize the input convolutional layer and output convolutional layer for the layer mask using the weights from the stable diffusion for images. We employ a learnable zero-initialized scalar parameter to control the mask information injection in to the layer image after the input convolutional layer.

## A.2 DATA CONSTRUCTION DETAILS

We use spaCy (Honnibal et al., 2020) to parse the phrases from a sentence for DeCLIP Yao et al. (2022). However, it is note that these phrases may include a series of unnecessary background descriptions such as 'the lighting' and 'white background'. To address this problem, we first implement a manual filtering process to preclude such superfluous elements, resulting in a refined collection of noun phrases, which we designate as layer prompts. Furhtermove, Manual filtering may not be enough to guarantee a perfect elimination of all background words, which may lead to subsequent detectors and SAM possibly introducing some mask attention to background areas such as the floor, which we desire to have in the background layer rather than the foreground layer. To alleviate this issue, we compare the masks obtained from SAM with those obtained from saliency detector, ICON (Zhuge et al., 2022), retaining the masks with an overlap rate greater than 10%. Simultaneously, to ensure the order of layers, we employ a depth estimation model, Midas (Ranftl et al., 2022)), to estimate the depth of the image, and obtain layer depth values by averaging the estimated depth values over the mask area. The layers are then arranged from deeper to shallower based on these depth values. During the Stable Diffusion inpainting process, an empty text description is used as the prompt, while the terms pertaining to foreground objects are utilized as negative prompts. However, as Stable Diffusion tends to inpaint a foreground object inadvertently, a saliency detection model (Zhuge et al., 2022) is employed to ascertain the presence of any foreground objects within the inpainted area, thereby enhancing the quality of the background layer.

## A.3 DATASET DETAILS

Our MLCID is a long-tailed dataset comprising 0.85M two-layer images, 0.2M three-layer images, and 0.07M four-layer images. The lack of a sufficiently extensive data results in the model lacking generalization capability in generating three-layer and four-layer images. In the future, we aim to further enhance our dataset construction efficiency and the quality of data construction, expand the scale, and improve the generative capability of our model.

## A.4 APPLICATION DETAILS

For layer inpainting, the timesteps for non-target layers are set directly to 1, and the information of non-target layers is not altered during the inference process. For the target layer, the layer latent is sampled directly from the Gaussian distribution. Similarly, we employ a 100-step DDIM scheduler. For layer style transfer, akin to the image2image approach, we introduce 25 steps noise to the target layer and perform denoising for 25 steps, while the timesteps for other layers remain at 1.

## A.5 MORE VISUALIZATION

In Figures 8 and 9, we provide more examples of generated multi-layered composable images, and compare them with the samples generated by Stable Diffusion, a whole-image generation method. This comparison aims to showcase the distinct capabilities and potential advantages of our layered generation approach over traditional whole-image generation techniques.

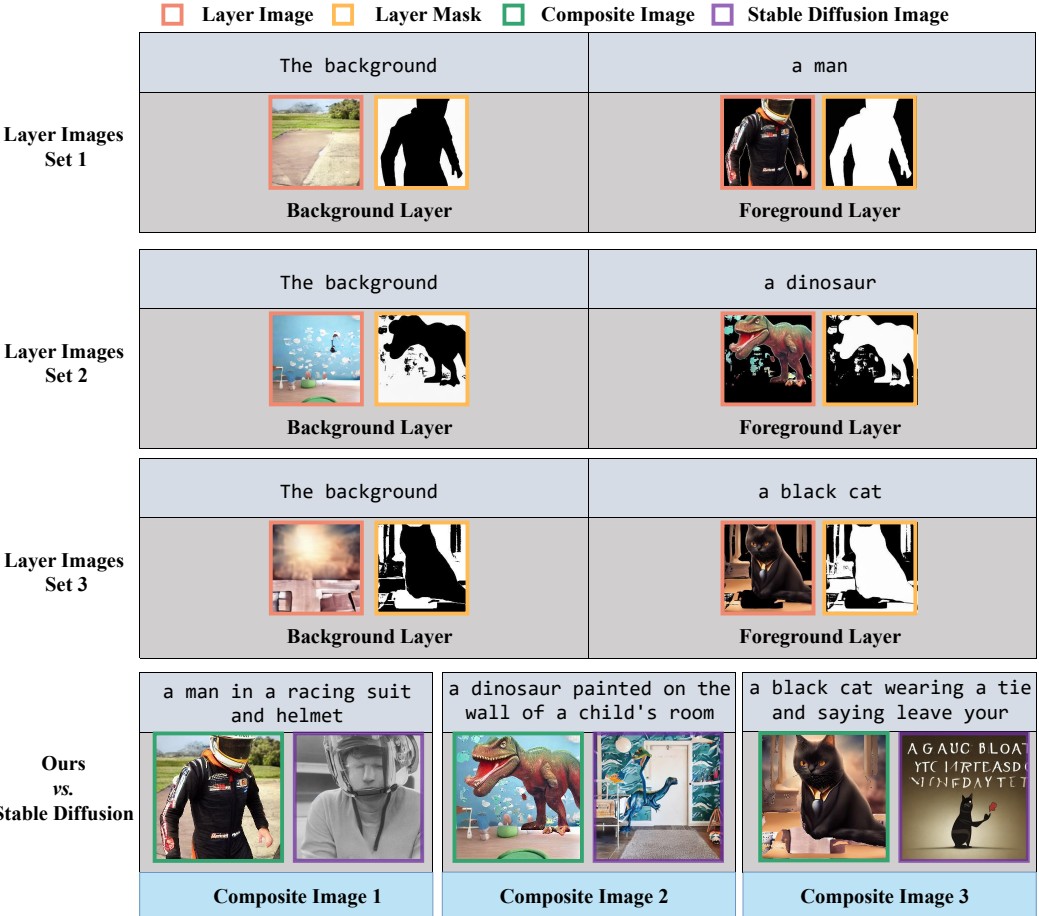

Figure 8: More visualization of the synthesized two-layered images.

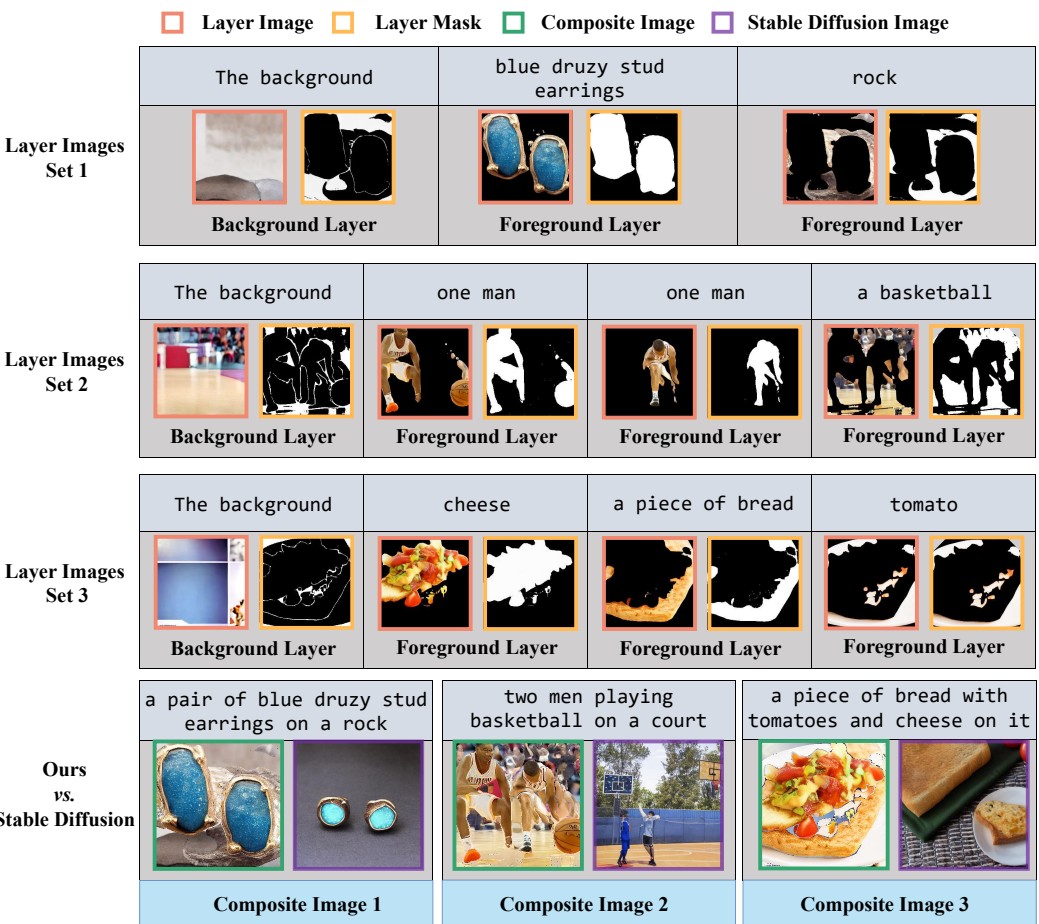

Figure 9: More visualization of the synthesized three-layered images and four-layered images.

