# OpenReview forum: "LayerDiff: Exploring Text-guided Multi-layered Composable Image Synthesis via Layer-Collaborative Diffusion Model"
_ICLR.cc/2024/Conference — ICLR 2024 Conference Withdrawn Submission_

### Official Review · Reviewer_4pq3 · 2023-10-29

**Soundness:** 2 fair
**Presentation:** 2 fair
**Contribution:** 2 fair
**Rating:** 3
**Confidence:** 4

**Summary:**

The authors propose a layer-based diffusion generation paradigm that enables multiple-layer image and mask generation in the diffusion process, leading to greater flexibility and control over the synthesized results. Specifically, an image is divided into one background layer and several foreground layers, each containing one foreground element. With a layer-specific prompt enhancer and a layer-collaborative diffusion model with specially designed inter-layer and intra-layer attention modules, the model is able to generate images and masks of multiple layers. A large-scale dataset with multi-layer images and layer-specific prompts is synthesized for training.

**Strengths:**

- Valid motivation
- Novel idea of layer-collaborative attention modules
- Well-designed data acquisition pipeline

**Weaknesses:**

- Inadequate experimental results: experimental comparison has only been made with the Stable Diffusion model, yet the results do not convincingly demonstrate the superiority of the proposed method.
- Missing comparison with relevant previous work: in Section 1, the authors reference [1] as a relevant previous work utilizing a layer-based synthesis approach. Though it is only capable of two-layered compositions, I think it is necessary to compare the two-layered results of the proposed method with this previous work.
- Missing ablation study on the Layer-Collaborative Attention Block.

[1 ]Xinyang Zhang, Wentian Zhao, Xin Lu, and Jeff Chien. Text2layer: Layered image generation using latent diffusion model. arXiv preprint arXiv:2307.09781, 2023.

**Questions:**

- Exceed page limit by two lines
- The main drawback of this paper is that the experimental results do not show the effectiveness of the proposed method. The authors might consider more challenging scenarios that align with the nature of layer-composable image synthesis like object-specific editing in multi-object images or complex compositions to demonstrate the superiority over Stable Diffusion.
- It is mentioned in section 3.1.1 that "There is no overlap between the foreground images and foreground masks, respectively". How is this guaranteed in the generation process? Also in 3.1.1, it says "Those foreground pairs are arranged along the depth information so that they can be stacked from top to bottom and overlaid onto the background image to compose an entire image." However, if the masks are non-overlapping, why does the order matter? And how to get such depth information?

---

### Official Review · Reviewer_YQTS · 2023-11-06

**Soundness:** 2 fair
**Presentation:** 1 poor
**Contribution:** 2 fair
**Rating:** 3
**Confidence:** 4

**Summary:**

This paper introduce a diffusion model to generate multi-layered composable images. In particular, the text prompt is parsed to layer-wise prompts, based on which the layer images and layer masks are synthesized respectively for each layer. A layer-specific prompt enhancer is introduced to parse the layer-wise prompts, and an inter-layer attention module is used to harmonize information across different layers.

**Strengths:**

1. LayerDiff offers a multi-layer approach towards composable image generation, which can decompose composable image generation to layer-wise tasks.
2. This paper demonstrated some qualitative results.
3. The multi-layer approach can enable some downstream tasks like layer-specific image editing.

**Weaknesses:**

1. The presentation is under satisfaction.
(1) Some minor mistakes. E.g., For Figure 1, in the second line of the caption, there is a missing space after period. E.g., in Page 7, in the second last paragraph, "DDIM Scheduler ()" has no content in the bracket.
(2) The total page count of the main paper exceeds 9 pages.
(3) The appendix is not submitted separately via supplementary, but as part of the main paper.
2. Some pipeline details are not elaborated clearly, please see my questions below.
3. Limited technical novelty: The overall framework seems more like an integration of multiple existing techniques. Although there are some original modules (e.g., inter-layer attention module), I still find their technical contributions relatively limited.

**Questions:**

1. Since layer masks are treated as RGB images, how do you binarize the predicted RGB masks to black-white binary masks?
2. At sampling stage, are the mask and corresponding image layer synthesized (i.e. denoised) simultaneously? If so, how do the noisy mask guide layer-specific synthesize at initial denoising stages, when the mask information is unclear?
3. For the inter-layer attention module, what is the input? For "layer hidden states" mentioned in Section 3.2.2, does it refer to intermediate noise output by the UNet? If so, how do the communication between cross-layer intermdiate noise help with the coherence and harmony across layers?
4. Could you provide experimental proof to showcase the usefulness of inter-layer attention module? For example, how can it help with enhanced coherence across layers.

---

### Official Review · Reviewer_RBAM · 2023-11-08

**Soundness:** 3 good
**Presentation:** 3 good
**Contribution:** 2 fair
**Rating:** 5
**Confidence:** 3

**Summary:**

This paper proposes LayerDiff, a Layer-collaborative diffusion model, to achieve text-guided multi-layered composable. The LayerDiff introduces a layer-based generation paradigm incorporating multiple layer-collaborative attention modules to capture inter-layer patterns. It also introduces a layer-specific prompt-enhanced module to capture detailed textual cues from the global prompt. Finally, LayerDiff presents a pipeline to obtain the dataset.

**Strengths:**

1. The layer-collaborative attention block is reasonable; this block can improve the intra-layer generation performance with a prompt enhancer and introduce the Interlayer attention module to enable cross-layer interaction.
2. The dataset has good value for the development of the community.

**Weaknesses:**

1. As shown in Fig.2, the multi-layer images and masks are generated simultaneously and combined to generate the final image with Eq.1, so there are obvious artifacts at the junction of different layers (see the Composite Image2 in Fig.9, Composite Image2 in Fig.5), has the author tried to solve this problem?

2. All results have a small resolution, making it difficult to see the superiority of this method. For example, I found the results of Stadble Diffusion in Composite Image2 have better performance.

**Questions:**

See the weakness.